# An unusual bird (Theropoda, Avialae) from the Early Cretaceous of Japan suggests complex evolutionary history of basal birds

Takuya Imai [1,2]*, Yoichi Azuma[1,2], Soichiro Kawabe [1,2], Masateru Shibata[1,2], Kazunori Miyata[2], Min Wang [3,4] & Zhonghe Zhou[3,4]

The Early Cretaceous basal birds were known largely from just two-dimensionally preserved specimens from north-eastern China (Jehol Biota), which has hindered our understanding of the early evolution of birds. Here, we present a three-dimensionally-preserved skeleton (FPDM-V-9769) of a basal bird from the Early Cretaceous of Fukui, central Japan. Unique features in the pygostyle and humerus allow the assignment of FPDM-V-9769 to a new taxon, *Fukuipteryx prima*. FPDM-V-9769 exhibits a set of features comparable to that of other basalmost birds including *Archaeopteryx*. Osteohistological analyses indicate that FPDM-V-9769 is subadult. Phylogenetic analyses resolve *F. prima* as a non-ornithothoracine avialan basal to *Jeholornis* and outgroup of the Pygostylia. This phylogenetic result may imply a complex evolutionary history of basal birds. To our knowledge, FPDM-V-9769 represents the first record of the Early Cretaceous non-ornithothoracine avialan outside of the Jehol Biota and increases our understanding of their diversity and distribution during the time.

[1] Institute of Dinosaur Research, Fukui Prefectural University, 4-1-1 Matsuoka Kenjojima, Eiheiji, Fukui 910-1195, Japan. [2] Fukui Prefectural Dinosaur Museum, 51-11 Terao, Muroko, Katsuyama, Fukui 911-8601, Japan. [3] Institute of Vertebrate Paleontology and Paleoanthropology, Chinese Academy of Sciences, 142 Xizhimenwai Street, 100044 Beijing, China. [4] Center for Excellence in Life and Paleoenvironment, Chinese Academy of Sciences, 142 Xizhimenwai Street, 100044 Beijing, China. *email: t-imai@dinosaur.pref.fukui.jp

Recently, our understanding of the evolution of birds during the Early Cretaceous has been dramatically improved by the exceptionally preserved fossils from north-eastern China (Jehol Biota[1]), north-western China (Mazongshan Fauna[2]), Spain (Las Hoyas Fauna[3]), and Brazil (Crato Fauna[4,5]). We now grasp their general diversification and course of evolution in the late Early Cretaceous. Well-known *Archaeopteryx* from the Upper Jurassic Solnhofen Limestone of southern Germany is the most basal taxon in the clade Avialae[6,7] (we conventionally define a common term birds corresponding to the Avialae in this paper[8], realizing the definition of the clade is debated). The Early Cretaceous Jeholorniformes is the next stemward avialan[9,10] and lacks the pygostyle, a compound element formed by a series of fused and shortened distal caudal vertebrae. More derived than Jeholorniformes are the clade Pygostylia, which includes birds with pygostyles. Basal pygostylians, including the Confuciusornithidae[11,12], Jinguofortisidae[13,14], and Sapeornithidae[15,16], exhibit pectoral and forelimb features that are less comparable to modern birds than those in the more derived clade Ornithothoraces. The Ornithothoraces includes Enantiornithes (extinct by the end of the Cretaceous)[17] and Ornithuromorpha[18] (in which modern birds are nested), and both of them are well-adapted to powered flight.

Specifically, multiple lineages of basal birds seem to have independently evolved flight-related morphological characters seen in extant volant birds. For example, while being the most basal avialans that is only surpassed by *Archaeopteryx*, the Jeholorniformes possesses a pectoral structure implying a capacity for powered flight[19]. The Confuciusornithidae is the most basal pygostylians that possess an ossified sternum for the attachment of large pectoral muscles, while they retain a scapulocoracoid (fused scapula and coracoid) as in non-avian theropods[20]. This condition is shared in the Jinguofortithidae, whilst their manual morphology is relatively derived (i.e., strongly curved minor metacarpal, and reduced minor digits)[14]. The Sapeornithidae bears particularly elongate forelimbs and separated scapula and coracoid, while they do not seem to possess an ossified sternum[16]. In Enantiornithes and Ornithuromorpha, the pectoral girdle is further refined with an ossified sternum with a large keel, strut-like coracoid, and a U- or V-shaped furcula, along with an advanced degree of bone fusion in the pelvis, forelimbs, and hindlimbs[17,18]. These observations demonstrate that basal birds performed extensive flight-related experiments on their skeletal structures during their early evolution.

Nearly all the Early Cretaceous non-ornithothoracine birds were known from the Jehol Biota of north-eastern China, the paleoenvironment of which is characterized by a highland lacustrine setting, temperate to relatively cold climate (estimated average temperature is $10 \pm 4\,°C$) and extensive volcanic activity in the surrounding area[1,21,22]. This palaeogeographic bias hinders not only our understanding of the global distribution of these most-basal clades but also of the evolution of flight-related features in different environments during this critical temporal span. In addition, nearly all the Early Cretaceous birds are two-dimensionally preserved, severely limiting the amount of morphologies that can be confidently reconstructed.

In 2013, the first associated skeleton of an Early Cretaceous bird from Japan was collected in Katsuyama, Fukui, central Japan. Unlike most other Early Cretaceous birds, the specimen is three-dimensionally preserved, and exhibits several autapomorphies, leading to erect a new taxon, *Fukuipteryx prima* gen. et sp. nov. Multiple morphological features indicate that *F. prima* is a non-ornithothoracine bird and is the first such record to our knowledge outside north-eastern China. Despite the fact that *F. prima* possesses a pygostyle, the phylogenetic analysis recovers it as a non-pygostylian avialan basal to *Jeholornis*. While this peculiar phylogenetic hypothesis may partly result from the subadult nature and incompleteness of the specimen, the study increases our understanding about the complex morphological evolution in early birds with the presence of particularly primitive features in young individuals.

## Results

### Systematic palaeontology

<div align="center">

Theropoda Marsh, 1881

Maniraptora Gauthier, 1986

Avialae Gauthier, 1986

*Fukuipteryx prima* gen. et sp. nov.

</div>

**Etymology**. Fukui refers to Fukui Prefecture in central Japan, where the specimen was collected, and pteryx (Latin) for wing; prima (Latin) for primitive, as the species exhibits several primitive morphological features among fossil birds.

**Holotype**. FPDM-V-9769 (FPDM: Fukui Prefectural Dinosaur Museum, Fig. 1), a disarticulated but closely associated skeleton, including right surangular; two cervicals; four dorsals including two nearly complete ones, an isolated centrum, and incomplete vertebral arch; incomplete synsacrum composed of two sacrals; five caudals; pygostyle; several dorsal ribs; incomplete furcula; incomplete left and right coracoids; incomplete right ilium; forelimbs including left humerus, incomplete left and complete right ulna, complete left and incomplete right radius, right major metacarpal, left minor metacarpal, left alular digit 1, left major digits 2 and 3, minor digits 3 and 4; and hindlimbs including left and right femur; left and right incomplete tibia; and variably preserved metatarsals II (left), III (incomplete left), and IV (complete left and incomplete right).

**Type locality and horizon**. Kitadani Dinosaur Quarry in the northern part of the city of Katsuyama, Fukui, Japan (Supplementary Fig. 1); the Lower Cretaceous Kitadani Formation (Aptian)[23] (see Supplementary Notes for further details).

**Diagnosis**. A pigeon-sized non-ornithothoracine avialan with the following autapomorphies: semicircular depression on the craniodorsal corner of the humeral head, dorsally bowed humeral shaft, and robust pygostyle with incipient spinal processes and paddle-like structure at the distal end.

**Description**. Most elements are preserved three-dimensionally. Nearly all cranial and mandibular elements are severely damaged, preventing identification, and many axial elements are missing. A series of caudal vertebrae are relatively well-preserved, and many appendicular elements are intact (Fig. 2).

The surangular bears a slender dorsal process that presumably forms the dorsolateral margin of the mandible (Fig. 3a). Caudally, the surangular is perforated by a circular foramen. While the surangular is incomplete and it is difficult to determine the rostral extent of the foramen, its position and circular shape are reminiscent of the caudal mandibular fenestra present in confuciusornithids[11,24–26] as well as in paravian *Anchiornis*[27] (described as a surangular foramen). While the surangular is mediolaterally compressed, it thickens to form a low hillock just caudal to the putative caudal mandibular fenestra and rostral to the articulation surface for the quadrate, as in confuciusornithids[26] (see Supplementary Notes for additional descriptions and comparisons of the surangular).

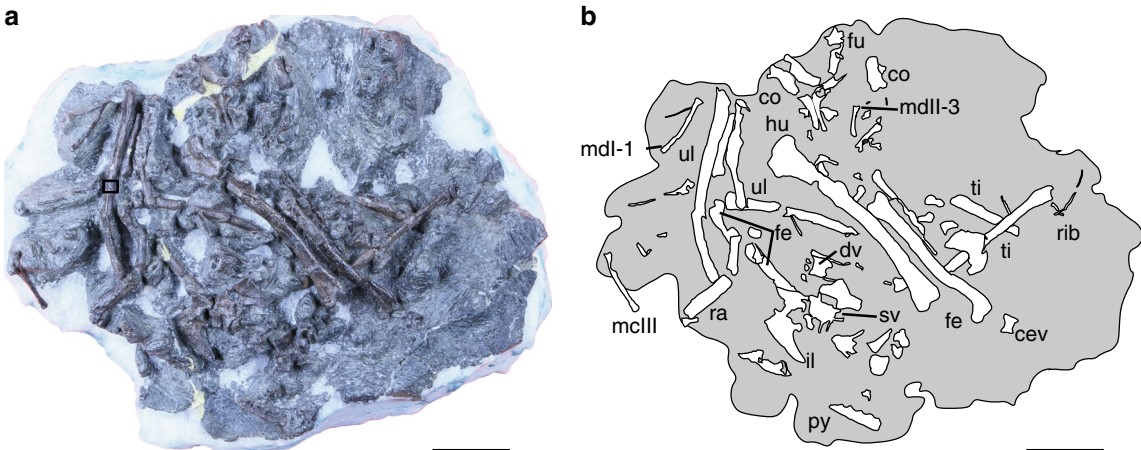

**Fig. 1 FPDM-V-9769. a** Photograph. A box indicates where a sample was taken for osteohistological analyses. **b** Schematic line drawing of the skeletons. Abbreviations: cev cervical vertebra, co coracoid, dv dorsal vertebra, fe femur, fu furcula, hu humerus, il ilium, mc metacarpal, md manual digit, py pygostyle, ra radius, sv sacral vertebra, ti tibia, ul ulna. Scale bars equal 3 cm in **a** and **b**

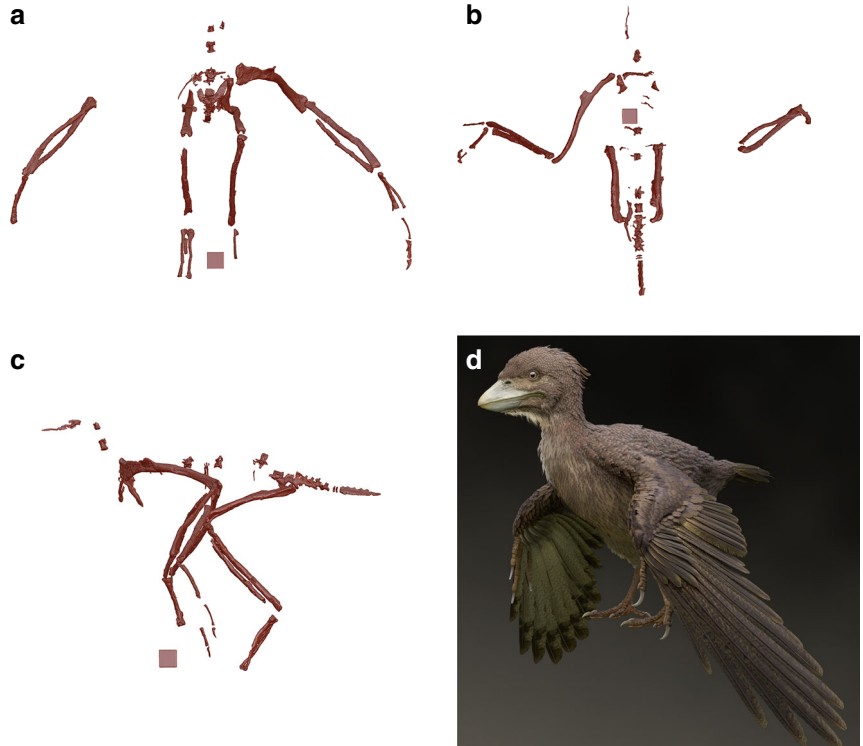

**Fig. 2 Reconstructions and restoration. a–c** Skeletal reconstructions of FPDM-V-9769 in cranial (**a**), dorsal (**b**), and left-lateral (**c**) views. **d** Life restoration of *Fukuipteryx prima* (artwork by M. Yoshida). One side of a cube in each image equals 1 cm

Two cervical vertebrae are recognized, although they are missing neural arches. The centra are craniocaudally longer than mediolaterally wide, and slightly dorsoventrally compressed caudally. Alternatively, the dorsoventral compression may be a preservational artefact (Fig. 3b). Their cranial articular surface exhibits incipient heterocoelous condition, whereas the caudal articular surface appears slightly concave. Similar observations have been reported in *Confuciusornis*[20] and some enantiornithines[28]. No pneumatic foramina are recognized on the centra as in *Jeholornis* but in contrast to *Archaeopteryx* and some non-avian theropods[29]. Ventral processes and costal processes are lacking.

The dorsal centra are procoelous with subcircular to subrectangular cranial articular surfaces, and relatively flat caudal

articular surface (Fig. 3c). The vertebral foramen is slightly dorsoventrally lower than the articular facet. Pneumatic fossae are present, comparable to those in *Confuciusornis*[20], *Sapeornis*, and enantiornithines[30]. Notably, an isolated dorsal centrum bears large and deep, suboval lateral excavations (Fig. 3d) as seen in *Confuciusornis*[20] but not in more stemward avialans such as *Jeholornis* and *Archaeopteryx*. In the two complete dorsal vertebrae, the neural spines are low cranially, moderately tall caudally, and craniocaudally wide. Presumably, these vertebrae represent cranial ones. In one incomplete dorsal vertebra lacking the centrum, the spinal process is tall and craniocaudally narrow, suggesting that it represents a caudal one. The transverse process is partially preserved in one dorsal vertebra and is slightly dorsally oriented with respect to the centrum.

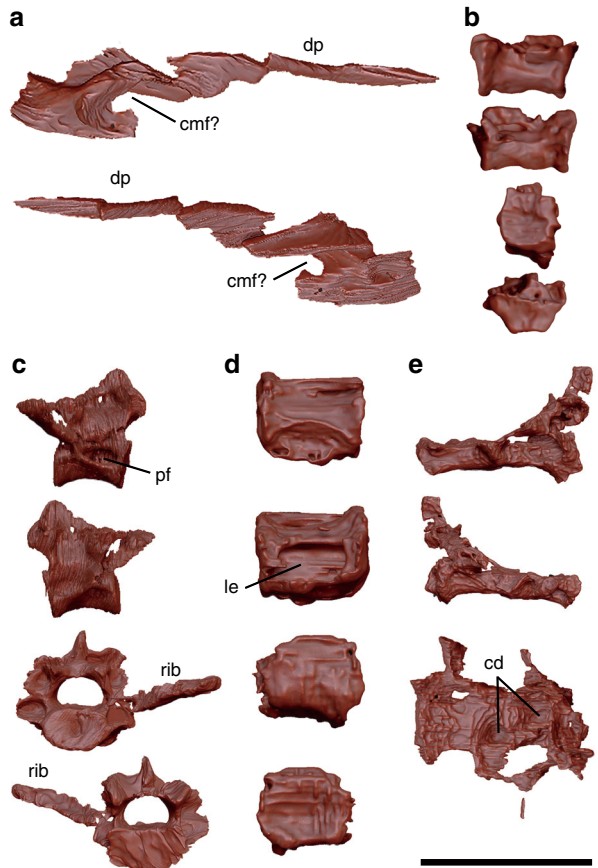

**Fig. 3 Cranial and selected axial elements except caudal vertebrae.**
**a** Surangular in medial and lateral views. **b** Cervical centrum in left lateral, right lateral, cranial, and caudal views from top to bottom. **c** Dorsal vertebra with incomplete left rib in left lateral, right lateral, cranial, and caudal views from top to bottom. **d** Dorsal centrum in left lateral, right lateral, cranial, and caudal views from top to bottom. **e** Synsacrum in left lateral, right lateral, and dorsal views from top to bottom. Abbreviations: cd circular depression, cmf caudal mandibular fenestra, dp dorsal process, le lateral excavation, pf pneumatic foramen. Scale bar equals 1 cm in **a–e**.

The synsacrum is incomplete and only preserves two fused sacral vertebrae and fragmentary neural spines (Fig. 3e). The centra are craniocaudally longer than they are dorsoventrally high, and the cranial and caudal articular facets are nearly flat. The ventral surfaces of the centra are concave, and a prominent lateral ridge is present on the caudoventral end of the caudal one. On the dorsal surface of the centra, two circular depressions are present; whether they are genuine features or preservational artefacts is unclear. The spinal process is low, and the vertebral foramen is large compared to the centra.

FPDM-V-9769 preserves five free caudal vertebrae; whereas seven are reported in *Confuciusornis*[20], five to six are estimated in *Jinguofortis*[14], seven to eight in *Sapeornis*[30], less than eight in enantiornithines[31], and six in the basalmost ornithuromorph *Archaeornithura*[18], suggesting *F. prima* possibly possesses six to eight free caudals. The centra of the more proximal free caudals exhibit suboval articular surfaces in which the cranial articular surface is slightly concave while the caudal articular surface is flat. In the most proximal caudal vertebra among the preserved elements, the neural spine is relatively tall dorsoventrally, but narrow craniocaudally with robust prezygapophyses (Fig. 4a). Two caudal vertebrae are articulated with each other. In these vertebrae, the posterior one bears the lower spinal process reduced to a notch. In addition, the postzygapophyses are

craniocaudally longer than the associated centrum. The preserved distalmost free caudal vertebrae are smaller than the preceding ones and lack the centra (Fig. 4b). They exhibit circular vertebral foramen and slender dorsolateral projections. Similar caudal vertebrae are reported in *Confuciusornis*[20], although the dorsolateral projections are longer in *F. prima*.

The pygostyle is present and ends with a paddle-like structure (ps in Fig. 4c) somewhat resembling that of the enantiornithine *Rapaxavis* in ventral view[32]. The pygostyle is long, robust, and rod-shaped as in confuciusornithids and enantiornithine longipterygids[33] (Supplementary Table 1). Incipient neural spines are distinguishable, indicating that it is composed of at least five vertebrae, whereas this number is eight to ten in *Confuciusornis*[20] and four in *Sapeornis*[30]. The pygostyle is triangular in transverse section and semi-rectangular in sagittal section, resembling that of *Confuciusornis*[33]. Intervertebral foramina are recognizable as shallow depressions ventral to the neural spines. The ventrolateral surfaces bear five small projections on both sides; because the number of these notches match that of the pygostyle-forming vertebra, these may be incipient transverse processes (tp? in Fig. 4c). The ventral surface is laterally lined by a pair of longitudinal ridges that resembles the ventrolateral processes seen in *Rapaxavis* and other longipterygids[32,33], forming a shallow furrow on the ventral surface. On the proximoventral surface of the furrow, a circular canal runs ventromedially into the pygostyle, and the computed tomography (CT) images show that this canal passes the ventral side of the pygostyle to the posterior end (Fig. 4d). The domestic chicken (*Gallus gallus domesticus*) exhibits a similar feature which houses the ventral tail vasculature including the aorta artery and caudal veins[34] (Fig. 4e). Thus, such features may have already been present in basalmost pygostylians as suggested in that of *F. prima*. While a vertebral foramen is obscured on the proximal articular surface, CT images reveal a canal bounded by thin walls within a vascularized internal pygostyle (Fig. 4d), resembling the spinal cord channel seen in *Confuciusornis* and the domestic chicken[34]. In contrast to adult chickens, where the spinal cord channel is craniocaudally short[34], that of *F. prima* seems to extend more distally. The anterior articular surface of the pygostyle is nearly flat, while the posterior articular surface is inclined to articulate with the distal paddle-like structure. This paddle-like structure is ridged along the midline, forming a triangular transverse section, and gradually becomes dorsoventrally thinner toward the distal end. This structure is found closely associated with the pygostyle and its anterior articular surface conforms with the posterior surface of the pygostyle. It was probably incompletely fused to the rest of the pygostyle and became separated during the preservation.

Five incomplete dorsal ribs are preserved, with one preserving a complete head (Fig. 5a). These ribs are slender, proximally curved but straight distally. Due to the state of preservation, it is unclear whether uncinate processes are present during life.

The furcula is robust and boomerang-shaped (Fig. 5b) as in other non-ornithothoracine birds except *Sapeornis*[13,14,20,35], and some derived non-avian theropods[27,36]. While its medial portion is damaged to some degree, the clavicular symphysis exhibits a smooth ventral margin, suggesting the absence of the hypocleidium. The interclavicular angle is estimated to be 70°, which is most comparable to that of *Chongmingia*[13]. Morphologically, the furcula of *F. prima* is most similar to that of *Confuciusornis*[20] in bearing a caudal projection with a rounded area at each omal end. As suggested for *Confuciusornis*, this area may be homologous to the acrocoracoidal articular facet of modern birds[20,37]. The furcula is in contrast to those of Ornithothoraces; ornithothoracine furcula is deep-U-shaped or V-shaped and slender. The coracoid is unfused with the scapula (Fig. 5c). Although the distomedial part is missing, the preserved portion of the coracoid

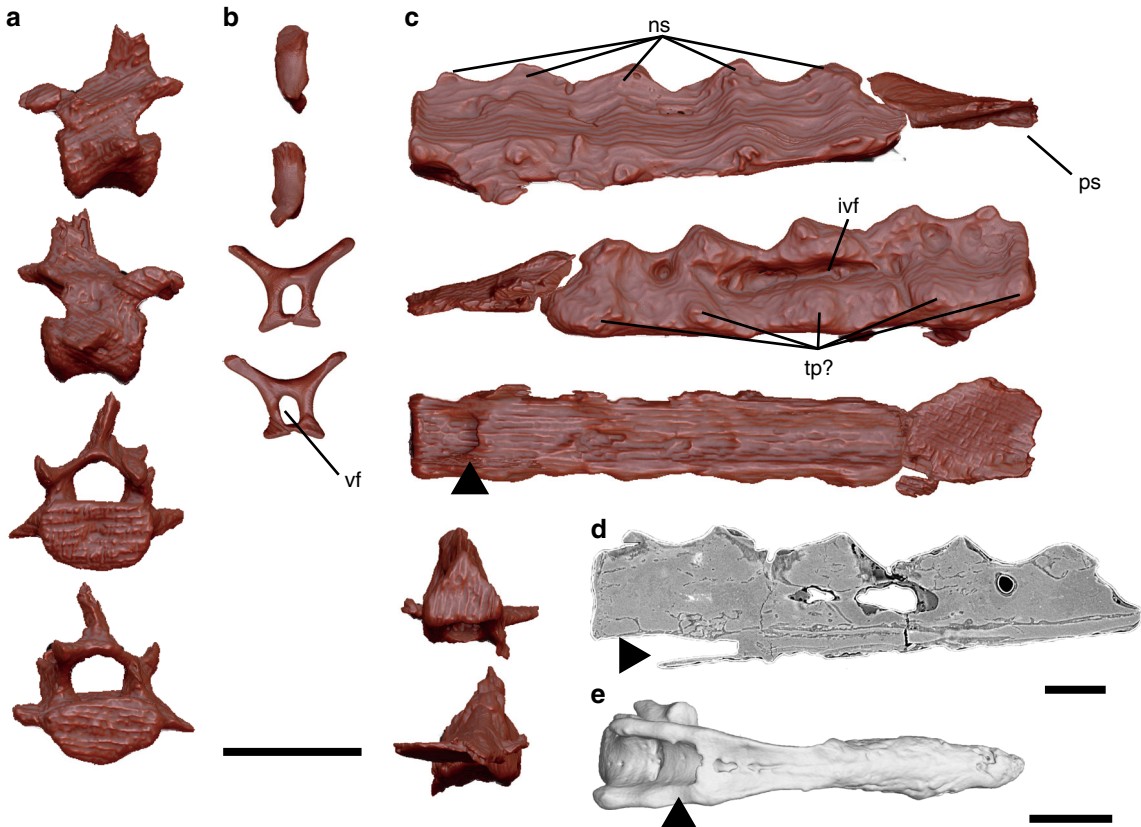

**Fig. 4** Caudal vertebrae. **a** Anterior caudal vertebra in left lateral, right lateral, cranial, and caudal views from top to bottom. **b** Posterior caudal vertebra in left lateral, right lateral, cranial, and caudal views from top to bottom. **c** Pygostyle in left lateral, right lateral, ventral, cranial, and caudal views from top to bottom. **d** Sagittal section of the pygostyle under computed tomography image. **e** Pygostyle of *Gallus gallus domesticus* in ventral view. Black arrows in **c**, **d**, and **e** denote ventromedial canals observed in FPDM-V-9769 and *G. gallus domesticus*. Abbreviations: ivf intervertebral foramen, ns neural spine, ps paddle-like structure, tp transverse process, vf vertebral foramen. Scale bars equal 5 mm in **a**–**c**, 2 mm in **d**, and 5 mm in **e**

appears strut-like and morphologically most similar to that of *Jeholornis*[9,10]. As in *Jeholornis*, the lateral process is well-developed and more elongate than in *Archaeopteryx* and *Sapeornis*. The scapular cotyle is large and the glenoid facet is robust. An ossified sternum is not present in FPDM-V-9769.

The incomplete ilium only preserves the acetabulum and postacetabular wing (Fig. 5d). Based on the preservation, the ilium (and the rest of the pelvic girdle) of FPDM-V-9769 is presumably unfused to each other nor are they fused to the synsacrum, as in other Early Cretaceous birds and most non-avian theropods[38]. The overall iliac shape resembles those of non-ornithothoracine birds and enantiornithines[38]. The postacetabular wing is dorsoventrally low, very mediolaterally compressed, and craniocaudally long. The pubic and ischial peduncles are robust and posteroventrally directed as in basal birds[38].

The humerus is longer than the femur (Supplementary Table 1), with a humerus/femur length ratio larger than those of *Archaeopteryx*, *Confuciusornis*, *Jinguofortis*, and *Chongmingia*[13,14] (Supplementary Table 1). The humeral shaft is bowed dorsally (Fig. 6a) unlike in most basal avialans. A wide and shallow capital incision separates the humeral head from the bicipital crest. The pneumotricipital fossa on the proximocaudal surface, a feature typical of many derived birds, is absent. Although the deltopectoral crest is largely incomplete, the preserved portion projects cranially, which is a derived feature absent in other non-ornithothoracine birds and enantiornithines. However, considering that the deltopectoral crest is severely damaged, it is equally likely that the cranial projection represents a preservational artefact. It is unknown whether the deltopectoral

crest is perforated as in confuciusornithids and *Sapeornis*. On the other hand, a semicircular depression is present on the craniodorsal corner of the humeral head and presumably proximal to the deltopectoral crest (scd in Fig. 6a). This feature has not been observed in other extant or extinct birds. The ulna is slender, weakly bowed caudally, and slightly shorter than the humerus (Fig. 6b; Supplementary Table 1), a condition shared with *Archaeopteryx* and confuciusornithids, but different from most volant birds[24]. The olecranon is poorly developed as in many other basal birds. No quill knobs for the attachment of the secondary remiges are recognized on the ulnar shaft. The radius is slightly bowed cranially (Fig. 6c). Its proximal end is cranio-caudally compressed, while the midshaft is cylindrical. The midshaft of the radius is nearly as robust as that of the ulna, in contrast to other non-ornithothoracines in which the ulna is substantially stouter[15,35]. The major and minor metacarpals are separated throughout their length. Among non-ornithothoracine avialans, this condition is only seen in *Archaeopteryx*[35]. The major metacarpal (Fig. 6d) is robust and slightly bowed dorsally, as seen in *Eoconfuciusornis*[11]. A small intermetacarpal tubercle is present on the caudodorsal surface of the shaft in *F. prima*, a feature absent in other non-ornithuromorph avialans but present in some extant birds[37]. The minor metacarpal, which appears more slender than the major metacarpal, is bowed caudally (Fig. 6e), but not to the extent seen in *Jeholornis* and *Chongmingia*[13]. The major metacarpal is longer than the minor metacarpal, a condition different from that of enantiornithines[17]. An intermetacarpal space is presumably present, while smaller than that of *Jeholornis* and *Chongmingia*[13]. The first phalanx of

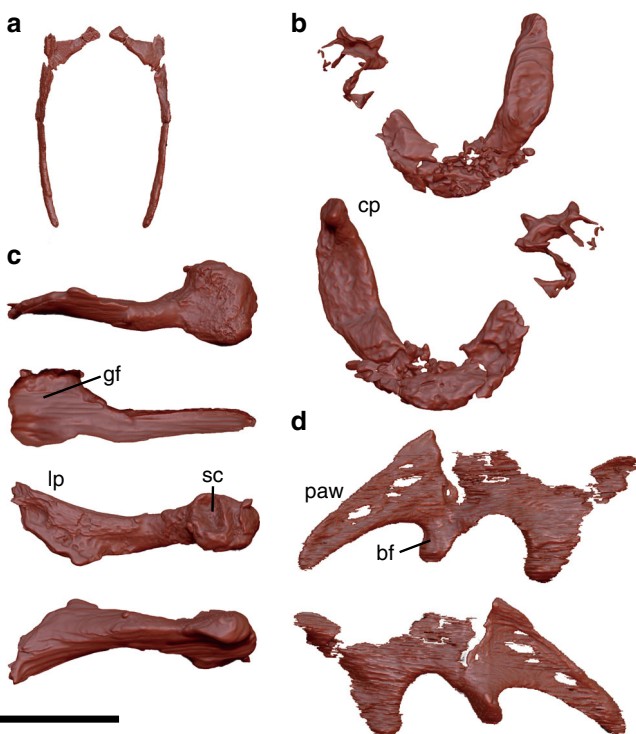

**Fig. 5** Rib, pectoral, and pelvic elements. **a** Right rib in cranial and caudal views. **b** Furcula in cranial and caudal views. **c** Left coracoid in medial, lateral, cranial, and caudal views from top to bottom. **d** Left ilium in medial and lateral views. Abbreviations: cp caudal projection, bf brevis fossa, gf glenoid fossa, lp lateral process, paw postacetabular wing, sc scapular cotyle. Scale bar equals 1 cm in **a**–**d**.

the alular digit (Fig. 6f) is slender and slightly bowed dorsally, as in *Eoconfuciusornis*[11]. This phalanx is craniocaudally flattened and tapers distally, rendering the distal end narrower than the proximal end. The second phalanx of the major digit is relatively more robust than other preserved phalanges and bowed caudally as in confuciusornithids[24] (Fig. 6g). The proximal and distal ends of this phalanx are expanded dorsoventrally, and the ventral surface is flat to slightly concave. The ungual phalanx of the major digit is slightly more than half the length of its preceding counterpart (Fig. 6h). The extensor and flexor tubercles are small. The third phalanx of the minor digit is slender and gently bowed dorsally (Fig. 6i). The ungual phalanx of the minor digit is incomplete, limiting the observation of morphological features (Fig. 6j). It is small and slightly curved with a blunt end, somewhat resembling that of *Archaeopteryx*[35].

The femora are relatively slender (Fig. 7a). They are bowed slightly laterally and cranially as in *Confuciusornis*[20] and *Sapeornis*[30] and bear a short neck that separates a round head from the trochanteric region, as in *Confuciusornis*[20]. The greater and lesser trochanters are fused to form a trochanteric crest that approaches to the proximal end of the femur, as in *Confuciusornis*[20] and *Chongmingia*[13]. The lateral and medial condyles are tall with deep intercondylar fossa between them. The tibiae are incomplete with the left tibia missing the distal end and the right tibia only preserving the midshaft (Fig. 7b). It is slightly bowed medially, possibly due to post-depositional deformation. The fibular crest extends 1/3 of the length, and a single cnemial crest is present. Among distal tarsals, the lateral-distal tarsal appears to cap the metatarsal IV (Fig. 7c). The metatarsals are unfused throughout their shafts, similar to *Archaeopteryx*[35]. Both metatarsals II and IV are slender, while the metatarsal II is

slightly longer than the metatarsal IV. The metatarsal IV bears a relatively larger proximal end than the metatarsal II.

**Osteohistological analysis**. To assess the ontogenetic stage of FPDM-V-9769, we conducted osteohistological analysis. A bone sample was taken from the midshaft of the left ulna following the standard method[39] (Fig. 1a). The ulnar transverse section appears vascular and is constituted predominantly by parallel-fibred bone tissue with longitudinal vascularization as in *Archaeopteryx* and *Jeholornis*[40] (Fig. 8a). Vascular canals are prominent and semi-circular to oval-shaped in most cases; occasionally transversely elongated canals are present (Fig. 8b). Vascularity decreases toward the periosteal region. It is noted that these canals are also visible in CT images of other long bones, such as the humerus, and exhibit a similar condition (Fig. 8c). The osteocyte lacunae are concentrated in the inner region and semi-circular in shape (Fig. 8b). Unlike *Archaeopteryx*, *Confuciusornis*, and *Sapeornis*, but resembling *Jeholornis*, the lacunae are not distinct nor densely distributed[40,41]. The medullary margin is uneven throughout the inner bone surface of the two transverse thin sections. No lines of arrested growth (LAGs) are recognized (Fig. 8a).

## Discussion

The following morphological features indicate *F. prima* is an avialan: robust pygostyle, robust U-shaped furcula with an interclavicular angle comparable to those of basal birds, strut-like coracoid unfused with scapula, humerus that is the most robust and conspicuous among appendicular elements, and an ulna and radius longer than the femur. Furthermore, as described above, *F. prima* exhibits numerous primitive features comparable to non-ornithothoracine birds but not to the more crownward taxa. These include: the U-shaped furcula without a hypocleidium as in *Archaeopteryx*, *Jeholornis*, confuciusornithids, and jinguo-fortithids (Fig. 5b), slender and relatively short ulna as in *Archaeopteryx* and confuciusornithids (Fig. 6b), and unfused major and minor metacarpals and metatarsals as in *Archae-opteryx* (Figs. 6d, e and 7c). On the other hand, *F. prima* lacks a series of characters shared by the more derived clade of avialans, namely Ornithothoraces. For example, the furcula is slender and deep U- or V-shaped in ornithothoracines, unlike that of *F. prima*. The pygostyle of *F. prima* is robust, rod-like in lateral view, and triangular-shaped in cranial view (Fig. 4c), while those of enantiornithine ornithothoracines are X-shaped in cranial view and those of ornithuromorph ornithothoracines are ploughshare-shaped and compact in lateral view[33]. The ulna is shorter than the humerus in *F. prima* in contrast to the proportionally longer ulna of volant ornithothoracines (Fig. 5a, b)[13]. Finally, despite that *F. prima* shares several skeletal features with enantiornithines, such as the rod-like pygostyle with paddle-shaped distal end and overall iliac shape, it is clearly different from all enantiornithine birds in having concave scapular cotyle (Fig. 5c), and major metacarpal longer than the minor metacarpal (Fig. 6d, e)[20]. These lines of evidence demonstrate that *F. prima* belongs to non-ornithothoracine avialans.

The absence of LAGs in the ulna of FPDM-V-9769 suggests the animal died when it was <1 year old. The presence of large vas-cular canals in the inner bone tissue suggests that the bird underwent a rapid phase of early development. However, the distribution of the vascular canals decreases in density toward the periosteal region, implying that the growth rate was declining prior to death. Given these observation, FPDM-V-9769 is likely a subadult nearly reaching its skeletal maturity at the time of death. This interpretation is further supported by the relatively medium body size (compared with other Mesozoic birds), and the variable degree of skeletal fusion in the preserved dorsal vertebrae, which

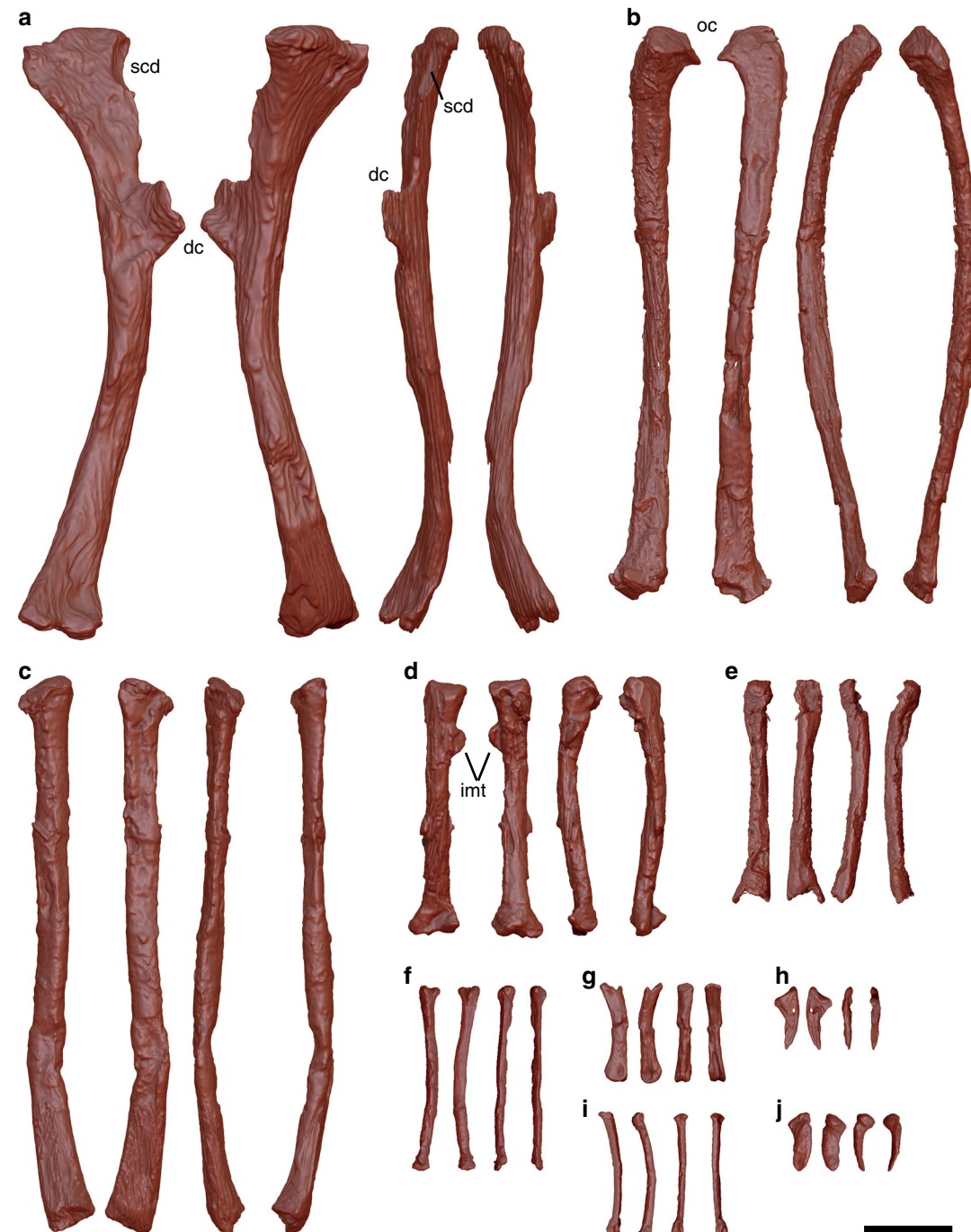

**Fig. 6** Forelimb elements. **a** Left humerus. **b** Right ulna. **c** Left radius. **d** Left major metacarpal. **e** Left minor metacarpal **f** Left alular digit 1. **g** Left major digit 2. **h** Left major digit 3. **i** Left minor digit 3. **j** Left minor digit 4. All elements are in cranial, caudal, dorsal, and ventral views from left to right in **a** through **c**, and dorsal, ventral, caudal, and cranial views from left to right in **d** through **j**. Abbreviations: dc deltopectoral crest, imt intermetacarpal tubercle, oc olecranon, scd semicircular depression. Scale bar equals 1 cm in **a**–**j**.

would be fully fused when the animal is completely matured[42]. The unfused metacarpals and metatarsals may further indicate the immature state of FPDM-V-9769.

Although skeletally immature, FPDM-V-9769 exhibits several skeletal morphologies that cannot be exclusively ascribed to ontogenetic variation. For example, the pygostyle is large, robust and well-fused, unlike 7- to 8-week-old chickens where the pygostyle is relatively slender and intervertebral space is discernible in CT images[34], suggesting the pygostyle of FPDM-V-9769 is nearly fully formed. A boomerang-shaped furcula is found in juvenile and subadult specimens of *Sapeornis*[43,44], and

the furcular shape does not seem to change dramatically during ontogeny in enantiornithines[31]. The ilium appears to be unfused to other pelvic elements in FPDM-V-9769, while the three pelvic bones are tightly fused in extant birds and some ornithothoracines[38]. Yet, they remain unfused in most other Early Cretaceous birds, particularly in all non-ornithothoracines, and separate pelvic elements do not necessarily reflect immaturity. Overall humeral shape also stays relatively the same in the growth series of *Sapeornis*[43,44] suggesting that the humerus of *F. prima* unlikely changes dramatically during later development.

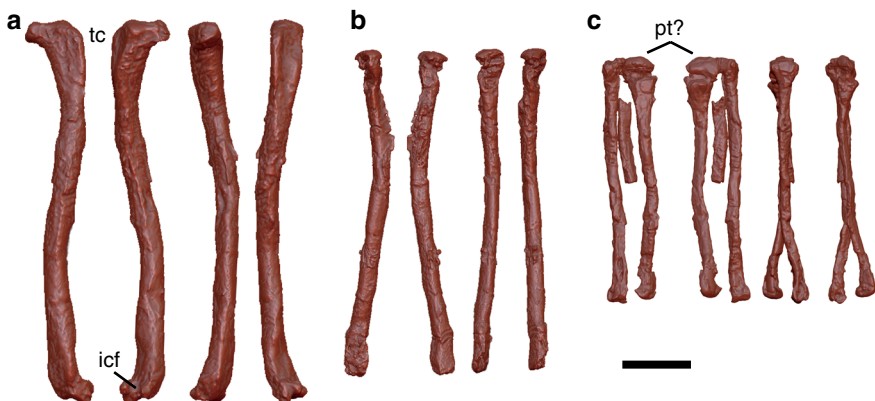

**Fig. 7** Hindlimb elements. **a** Left femur. **b** Right tibia. **c** Left metatarsals. All elements are in cranial, caudal, medial, and lateral views from left to right. Abbreviations: cc cnemial crest, dt distal tarsals, fc fibular crest, icf intercondylar fossa, tc trochanter crest. Scale bar equals 1 cm in **a–c**

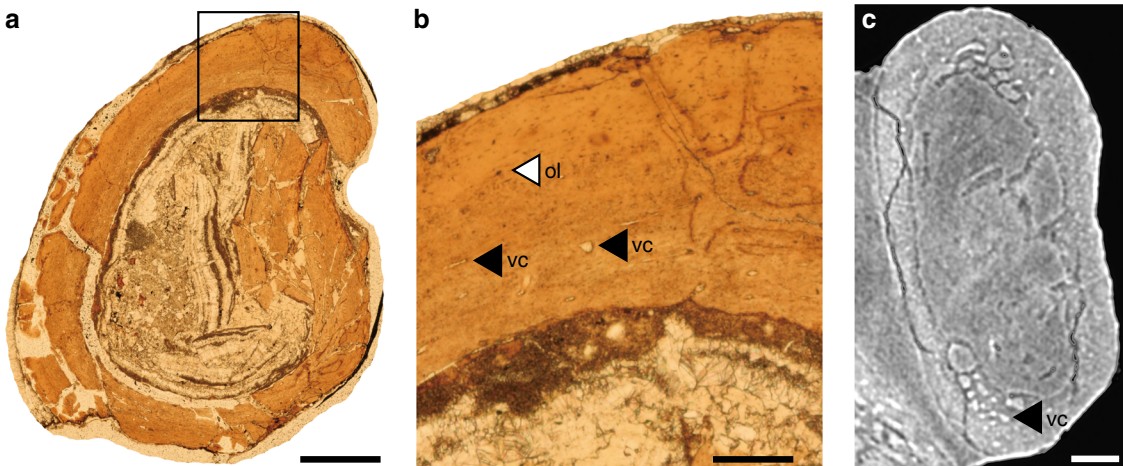

**Fig. 8** Osteohistological sections. **a** Transverse thin section of left ulna. Note absence of the lines of arrested growth. **b** Enlarged view of the portion of the thin section (box in **a**). Note round and elongated vascular canals (vc), and relatively scarce osteocyte lacunae (ol). **c** Transverse computed tomographic image of humerus. As in **b**, round and elongated vascular canals are present. Scale bars equal 5 mm in **a**, 1 mm in **b**, and 1 cm in **c**

To further explore the phylogenetic position of *F. prima*, we added FPDM-V-9769 to the most recent dataset targeting the phylogeny of Mesozoic birds[14]. In coding the character states for *F. prima*, any characters that are likely affected by the immaturity of the specimen were coded missing. These characters included Char. 64 (number of ankylosed sacral vertebrae), Char. 79 (absence or presence of ossified uncinate processes), Char. 108 (degree of ossification of the sternum), Char. 154 (degree of fusion between the semilunate carpal and metacarpals), Char. 166 (fusion of distal end of the metacarpals), Char. 178 (fusion of pelvic elements), Char. 209 (fusion between the Tibia, calcaneum, and astragalus), Char. 221 (fusion of distal tarsals with metatarsals), and Char. 248 (fusion between scapula and coracoid). Our phylogenetic analyses produced 1264 most parsimonious trees (MPTs) with a length of 1323. The strict consensus tree is fairly resolved, placing *F. prima* within the Avialae and outside of the clade Ornithothoraces (Fig. 9 and Supplementary Fig. 2). Bremer values are relatively low, reflecting missing characters in *F. prima* and other taxa. Interestingly, *F. prima* is resolved basal to the long-tailed *Jeholornis*, as the outgroup of the clade Pygostylia, despite that *F. prima* possesses a pygostyle. This may result from a series of primitive features found in *F. prima* such as simple and robust furcula, unfused pelvis, metacarpals and metatarsals, as well as a well-developed deltopectoral crest on the humerus extending to the midshaft. These features are comparable to those found in *Archaeopteryx*[35]

and *Jeholornis*[9,10]. Thus, the current phylogenetic position of *F. prima* may indicate a more complex evolutionary pattern in the body plan of early birds, as demonstrated in another potentially-non-pygostylian bird *Chongmingia*[13]. For example, pygostyle-like structures are found in some non-avian theropods[45–47]. Additionally, whereas the pygostyle has been suggested as one of the key flight adaptations in the early evolution of birds[23], a recent review argues that the early pygostyle is merely a by-product of the tail reduction to the extreme and unrelated to the flight adaptation[10]. In the present phylogenetic hypothesis, the fusion of distal caudal vertebrae (Char. 69 in Supplementary Notes) is excluded from the set of synapomorphies for a clade comprising the Confuciusornithidae, Jinguofortithidae, *Sapeornis*, and Ornithothoraces (Supplementary Fig. 3), the taxa that share the pygostyle and form the clade Pygostylia in previous analyses[11–19]. Given that the pygostyle is present in *F. prima*, a taxon basal to long-tailed *Jeholornis*, further tests may be necessary to understand the phylogenetic implications of the presence of the pygostyle in the Avialae.

Except for the Late Jurassic *Archaeopteryx*, non-ornithothoracine birds had previously been known only from the Jehol Biota and contemporary deposits in northern Korean Peninsula. The discovery of *F. prima* further increases the geological distribution of non-ornithothoracine birds. It appears that non-ornithothoracine avialans are not restricted to a relatively cold, highland lacustrine environment in the Early Cretaceous of

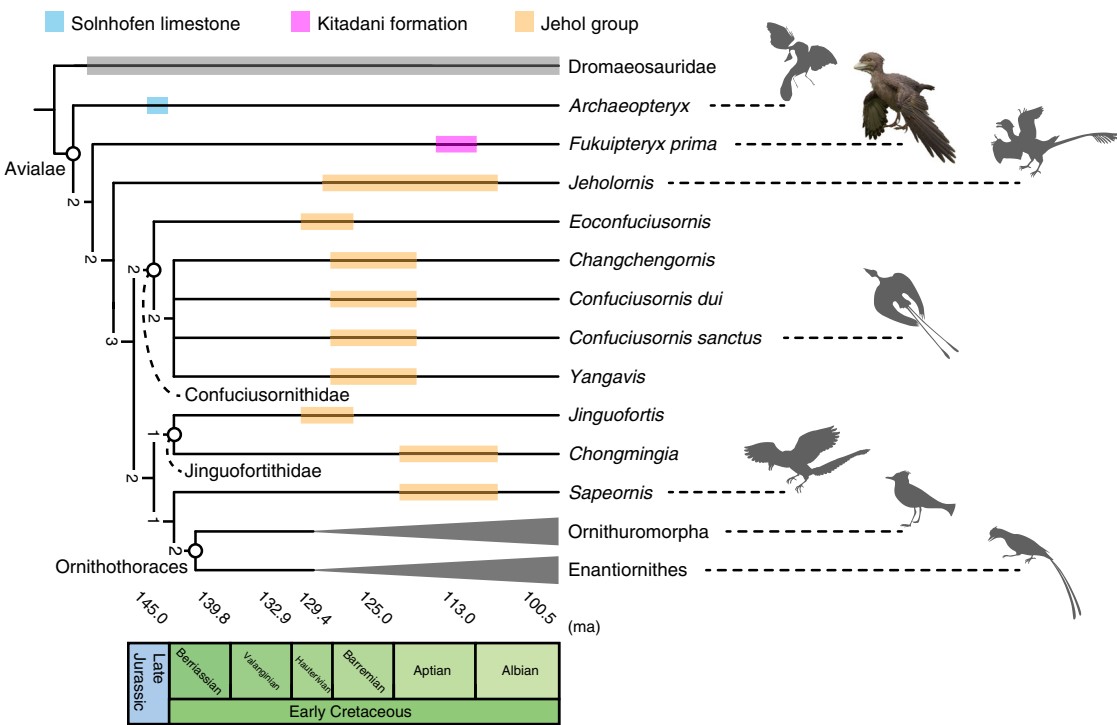

**Fig. 9** Simplified strict consensus tree of 1264 most parsimonious trees with a length of 1323, resulting from analyses with an addition of FPDM-V-9769 to a matrix on fossil birds[16]. Numbers near each node denote bremer support values. Note that *Fukuipteryx prima* represents the only Early Cretaceous non-ornithothoracine avialan collected outside of the Jehol Group. Silhouette drawings for Mesozoic birds were created by M.W. Life restoration of *F. prima* was created by M. Yoshida

north-eastern China, but inhabited more temperate, lowland regions such as the one represented by the Kitadani Formation, most likely with other ornithothoracines widespread around the globe. Further exploration of the Early Cretaceous fossil birds outside East Asia are greatly needed to clarify the palaeogeographical distribution of these basal birds. Very recently, several new genera and species of non-ornithothoracine avialans have been described from north-eastern China[10,13,14,24]. These discoveries and this study suggest non-ornithothoracine avialans may represent a diverse group within the Avialae. The evolution of flight-related apparatus may be more complex than previously thought, as illustrated by the mosaic assembly of non-pygostylian avian morphology and the very presence of a pygostyle in *F. prima*. Overall, this study demonstrates that further exploratory efforts in the Early Cretaceous sediments, where the occurrence of avian skeletons is limited to none, are important to increase our understanding of the Early Cretaceous birds.

## Methods

**CT experiments**. We employed a non-destructive anatomical analysis with a phase-contrast synchrotron micro-CT technique. The CT experiments were conducted at BL20-B2 of SPring-8 (RIKEN/JASRI), Sayo, Hyogo, Japan, with the following acquisition parameters: energy = 70 keV or 113 keV, number of projections = 900 per 180°, propagation distance = 4.2 m, exposure time = 1.0–3.0 s per projection, and pixel size = 12.92–24.1 µm. The tomographic data were converted into 16-bit.tif format and skeletal elements were segmented and rendered with VGStudio 3.1 and Amira 6.2.

**Phylogenetic analysis**. To assess the phylogenetic position of *F. prima*, we added FPDM-V-9769 to the most recent dataset on Mesozoic fossil birds[16] (Supplementary Notes). The dataset consisted of 280 morphological characters and 71 taxa. We employed TNT v. 1.5 (ref. [48]) with the character ordering same as in ref. [16], all the characters equally weighted, and 35 characters ordered (Supplementary Notes). The analysis was performed with the default setting except for the following: maximum number for trees in memory = equal to 10,000. We used the traditional search to search the MPTs with tree bisection-reconnection algorithm (TBR) where 1000 replicates of random stepwise addition and 10 trees held at each

step, 1000 replicates, and 10 trees per replicate. We then conducted a second TBR search using the MPT's obtained in the previous procedure where branches were collapsed if the minimal branch length was zero. The strict consensus tree was generated based on the MPT's found after the second TBR search. Bremer values were calculated as the support indices.

**Nomenclatural acts**. This published work and the nomenclatural acts it contains have been registered in ZooBank, the proposed online registration system for the International Code of Zoological Nomenclature (ICZN). The ZooBank LSIDs (Life Science Identifiers) can be resolved and the associated information viewed through any standard web browser by appending the LSIDs to the prefix http://zoobank. org/. The LSIDs for this publication are: EF35425E-B0AB-4950-BBA9-F86595BE7094 for the new genus *Fukuipteryx*, and D7369E30-E95A-415F-8B3E-894BF91F73F7 for the new species *F. prima*.

**Reporting summary**. Further information on research design is available in the Nature Research Reporting Summary linked to this article.

## Data availability

Morphological character descriptions and character datasets for phylogenetic analyses are included as Supplementary Notes. The FPDM-V-9769 specimen is housed at the Fukui Prefectural Dinosaur Museum. The rendered three-dimensional model for FPDM-V-9769 is available at https://doi.org/10.6084/m9.figshare.9869537. Requests for materials can be directed to the corresponding author.

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

## Acknowledgements
The authors thank staffs and excavation crews at FPDM for helping excavation and specimen preparation. Among the colleagues at FPDM, we express our gratitude particularly to T. Sekiya who assisted the phylogenetic analysis, H. Yukawa who shared his knowledge on the geological setting, and S. Hattori who helped reconstruction of the skeleton. The Japan Synchrotron Radiation Research Institute (JASRI) approved our CT experiments at the BL20-B2 of SPring-8 (Proposal Numbers 2017A1708 and 2017B1756), to which we are grateful. K. Uesugi and M. Hoshino at JASRI assisted the experiments during our work at SPring-8. Y. Deyama, M. Nakanishi, and Y. Nakamura at Fukui Prefectural University assisted the segmentation of CT images. M. Yoshida at Kobe Design University created skeletal reconstruction and life restoration of the specimen. We are grateful to Y. O'Connor for correcting English of the manuscript. The research was funded by Prefectural Government of Fukui (28-11).

## Author contributions
T.I and Y.A. designed the project and performed the research, T.I., S.K. and K.M. conducted the CT experiments, M.S. supervised the excavation and initial preparation of the specimen, and T.I., M.W. and Z.Z. wrote the manuscript.

## Competing interests
The authors declare no competing interests.
