## [Peer Review File · Communications Biology]

Reviewers' comments:

Reviewer #1 (Remarks to the Author):

This study reports a new taxon of Early Cretaceous avialan (hereafter 'basal bird', following the authors' usage) from Japan.

The skeleton is reasonably complete, although the images suggest that the specimen is not very well preserved and is in some ways difficult to interpret. It has been subjected to CT scanning so the skeletal elements are figured and described individually, although the resultant volumes are somewhat low resolution and not particularly straightforward to interpret.

Perhaps most significantly, the specimen represents the first basal bird from the Early Cretaceous of Japan, and the first such discovery from outside of China, Korea, or Germany.

The description of the skeletal material is relatively thorough, although it would be helpful for more information about how certain elements were identified. For example, what makes the authors sure that the putative surangular is in fact a surangular? The volume presented is difficult for me to assess.

The authors perform a phylogenetic analysis, although due to missing data support values for the topology are weak, and there is uncertainty as to whether some of the osteological interpretation is confounded by the skeleton deriving from a subadult animal, as suggested by an osteohistological assessment. It would be good if this paper could also be reviewed by someone with expertise in this kind of osteohistological analysis, which I have little experience with.

I feel that everything I've written above makes for a straightforward, if not especially high-profile, study. Unfortunately, the manuscript does suffer from editorial problems throughout—there are numerous spelling errors, and several sentences don't make sense. I do not want to hold the quality of the language in the paper against the authors, but it is clear that the paper will need to be heavily revised in order for it to be clearly understandable by most readers. I have noted a small number of such issues below, as well as other minor comments.

Line 50: Ornithuromorpha is misspelled

Line 50: The statement about enantiornithines going extinct at the end of the Cretaceous should be supported by a reference

Lines 53-55: This sentence doesn't make sense as written--I think 'suppressed' is not the right word here?

Description of holotype: What is the difference between 'incomplete' and 'partial'?

Description: The authors frequently use the term 'skeleton' instead of 'bone' or 'element'

Reviewer #2 (Remarks to the Author):

Review of "A new and bizarre basal bird (Theropoda, Avialae) from the Early Cretaceous of Japan", by Imai and co-authors.

I am happy to know about the discovery of this new and remarkable Cretaceous avian from Japan. Congratulations to the authors for such an interesting discovery, constituting one of the very few preserved three-dimensionally. The preservation is very good and affords valuable anatomical information not only allowing support the validity of the new taxon, but also to discuss evolutionary aspects of early birds as a whole. The high quality images they offer are quite useful to know the osteology of these basal birds. No doubt, the manuscript deserves publication in *Communication Biology*. Although I have enjoyed by reading the provocative conclusions expressed in the ms, I am not entirely convinced of them.

General considerations about the ms are as follows:

1. The anatomical descriptions are perfect, but I recommend expanding a little bit the universe of comparisons, including other basal birds (particularly pygostilians). Thus, the paper requires more explanations to justify why Fukuipteryx falls outside Ornithothoraces, instead of being a basal member of Pygostilia.

2. The ms begins with a conclusion: convergence was a common topic among basal birds, and Fukuipteryx represents one of these cases. I am not saying that convergence was not present in this part of the tree of life, but I believe the allocation of Fukuipteryx outside Pygostilia seems counterintuitive. It is hard to believe that such a "modern" kind of pygostile present in Fukuipteryx results in a case of convergence with that of derived pygostilians (e.g., enantiornithes).

3. Although I guess that Fukuipteryx is a pygostilian, I agree with the authors that it does not associate immediately with any of the currently recognized basal pygostilian clades (i.e., Confuciusornithidae, Sapeornithidae, Enantiornithes). Thus, this is the reason I encourage the authors to review their original interpretations, but also considering the possibility that Fukuipteryx represents a pygostilian not particularly related with any of the currently recognized clades.

Particular observations on the ms are as follows:

ABSTRACT

I recommend modifying the text, by almost repeating what it is said in the main text about that early Cretaceous birds "are currently known from the two-dimensionally-preserved specimens from north-eastern China, Las Hoyas (Spain), and Araripe (NE Brazil)". Current knowledge on basal birds does not restrict to China alone.

DIAGNOSIS

With the aim to quickly realize the size of the specimen, I suggest comparison with a living bird. For example: "A non-ornithothoracine bird the size of a sparrow/chicken/turkey..." (choose the correct one).

Lines 122-123 – Authors cite Anchiornis as a "non-avian theropod". However, different recent papers (Agnolín and Novas, 2013; Agnolín et al., 2019; Pei et al. 2018) demonstrated the avian affiliation of Anchiornis, as originally proposed by Xu (2011).

Lines 137-138 – Authors say that dorsal centrum of Fukuipteryx bears large and deep, suboval lateral excavations, as in Confuciusornis, but not in more stemward avialans such as Jeholornis and Archaeopteryx. Important is to also include that enantiornithes also bear such large excavation on dorsal centra.

Lines 152-155 – Please, insert information about the number of free caudals in enantiornithes and basal ornithurines.

Lines 152-155 – Why do you assume Fukuipteryx had more than five free caudals?

Lines 176 – 178 – Similarities that Fukuipteryx shares with some enantiornithes in the particular shape of pygostyle, can be used in support that Fukuipteryx is a bird related with enantiornithes, rather than consider these features as convergent acquisitions.

Lines 183-184 – Authors say "Thus, such feature may have been already present in basalmost birds with pygostyle as suggested in that of *F. prima*". I suggest to change this phrase for "Thus, such feature may have been already present in basalmost pygostilians as suggested in that of *F. prima*".

DISCUSSION

Ideas developed in present "Discussion" are based on the assumption that Fukuipteryx is an avialan less derived than Jeholornis. However, Fukuipteryx exhibits several enantiornithine/ornithothoracine features suggesting a higher position among birds.

Line 292 – Authors say "The following morphological features indicate Fukuipteryx prima is an avialan". However, Fukuipteryx exhibits features that are clearly MORE DERIVED than basal avialans (e.g., Archaeopteryx, Jeholornis, Rahonavis).

Lines 296-297 – Authors say "Furthermore, *F. prima* exhibits numerous primitive features comparable to non-ornithothoracine birds". Please, list these features.

Lines 308-309 – Authors say that Fukuipteryx "exhibits several primitive features that cannot be explained solely by ontogeny. For example, the pygostyle is large, robust and well-fused". Nevertheless, this kind of pygostyle gathers a set of clearly derived features, being morphologically far from a primitive condition, as represented by Sapeornithidae and Confuciusornithidae.

Line 314 – Another feature considered by authors as "primitive" for Fukuipteryx is pelvic elements unfused each other. But this is an ontogeny-dependent feature, and cannot be taken as indicative for a

basal position of Fukuapteryx among Avialae.

Lines 325-327 - The strict consensus tree supports the basal position of Fukuapteryx prima within Avialae and outside of Ornithothoraces. I highly respect authors for cladistics analysis they did, but the placement of Fukuapteryx as a bird less-derived than Jeholornis seems counterintuitive. General aspect of each of the available bones of Fukuapteryx (particularly the complex shape of the pygostyle) suggests it as a pygostilian bird. I recommend to discuss this point, as well as to make some comments if it is closer to Enantiornithes or Jinguofortisidae.

Lines 341-343 - Authors state that "skeletal features of F. prima demonstrate that the shortening and fusion of caudal vertebrae are not necessarily associated with the flight adaptations". Authors are right about that the origin of a pygostyle does not necessarily had an univocal relation with flying capabilities. However, and regarding Fukuapteryx, I am unable to recognize which are those "skeletal features" which are not related with flight. On the contrary, all the wing bones of Fukuapteryx closely resemble those of a flying bird.

Line 348 - Authors defend that the pygostyle evolved independently from flight capabilities. However, the kind of pygostyle present in Fukuapteryx seems more derived than that of Confuciusornis, resembling that of enantiornithes. A "pygostyle" (that is: fusion of distal caudals) evolved in oviraptorosaurs and pygostilian birds, and may have evolved many times among basal birds. However, a pygostyle with the system of crests and grooves present in Fukuapteryx gathers much more characteristics than "fused vertebrae" alone. I am unable to say that such a complex pygostyle IS NOT related with flight control.

Lines 352-353 - Authors refer about that "The discovery of Fukuapteryx prima further increases the geological distribution of non-ornithothoracine birds." But if Fukuiraptor, instead, is a basal pygostilian, then this paleobiogeographic conclusion could be incorrect. Accepting Fukuapteryx as a member either of Enantiornithes or Ornithothoraces, then the paleobiogeographic conclusion results more congruent with already known pygostilian/ornithothoracine temporal and paleobiogeographic distributions.

ILLUSTRATIONS

The illustrations of the present ms are excellent and I hope they become fully published.

I have inserted some observations close to some of the illustrated bones. By observing them, it seems clear that Fukuapteryx (even accepting it is a pygostilian), it does not comfortably fit among enantiornithes, for example. Thus, although I do not share author's results depicting Fukuapteryx as a non-ornithothoracine bird, I agree with them in viewing Fukuapteryx as a bizarre pygostilian!

Notes on the figures are as follows:

Figure 3b. The well-developed latero-dorsal crests conform a derived character, absent in basal pygostilians, but present in enantiornithines. This view is almost identical to Mirarce (Atterholt et al., 2018).

Figure 4b. The furcula is robust, but different from enantiornithes in lacking a hypocleidium.

Figure 4c. The acromial process of coracoid does not surpass the cranial level of coracoid, similar to basal birds and different from derived ornithothoracines.

Figure 4c. The coracoid exhibits a modern aspect in being strut-like, different from most basal birds. Thus, it fits well among derived pygostilians. However, it differs from enantiornithines in the following features: 1) proximal end is robust, apparently different from more slender one of enantiornithes; 2) the glenoid is CONCAVE, different from the flat or convex one of enantiornithes; 3) the dorsal surface of coracoid seems devoid of the typical excavation of enantiornithes.

Figure 5a. The hole present on the proximo-medial corner of humerus, is it an artifact of preservation? How it relates with the broken proximal half of deltopectoral crest?

Figure 5b. This extensive and rectangular deltopectoral crest is also present among enantiornithes.

Figure 5c. Could you offer information if an interosseous groove - a characteristic of many enantiornithes- is present or not on the radius?

Figure 5d. The abbreviation "imt" is not explained in figure text.

Figure 5. A reconstruction of the entire manus will be welcome!

Figure 6a. The proximal end of femur reminds me that of El Brete (Argentina) enantiornithes.

Figure 6b. The apparent lack of cnemial crest in proximal tibia resembles enantiornithes.

Figure 8. Take note of the spelling of "Ornithuromorpha"

Figure 9b. I suggest removing the life restoration of Fukuapteryx, thus gaining space for an informative anatomical illustration depicting, for example, bony proportions in both side and dorsal views.

Fernando E. Novas

Reviewer #1 (Remarks to the Author):

This study reports a new taxon of Early Cretaceous avialan (hereafter 'basal bird', following the authors' usage) from Japan.

The skeleton is reasonably complete, although the images suggest that the specimen is not very well preserved and is in some ways difficult to interpret. It has been subjected to CT scanning so the skeletal elements are figured and described individually, although the resultant volumes are somewhat low resolution and not particularly straightforward to interpret.

We consider that the preservation of the specimen is reasonably good, as noted by Reviewer 2. While the specimen does not preserve the entire skeleton, it is one of few examples of the Early Cretaceous birds that preserves substantial amount of bones three-dimensionally. We replaced some volumes with higher-resolution images. The images in the original manuscript was in low resolution for initial submission purpose. We are prepared to provide high-quality images upon request.

Perhaps most significantly, the specimen represents the first basal bird from the Early Cretaceous of Japan, and the first such discovery from outside of China, Korea, or Germany.

The description of the skeletal material is relatively thorough, although it would be helpful for more information about how certain elements were identified. For example, what makes the authors sure that the putative surangular is in fact a surangular? The volume presented is difficult for me to assess.

We provided additional notes on the identification of the surangular and comparison with that of confuciusornithids (Lines 124-126).

The authors perform a phylogenetic analysis, although due to missing data support values for the topology are weak, and there is uncertainty as to whether some of the osteological interpretation is confounded by the skeleton deriving from a subadult animal, as suggested by an osteohistological assessment. It would be good if this paper could also be reviewed by someone with expertise in this kind of osteohistological analysis, which I have little experience with.

We accept that the support values are low with the data available to us. In the original manuscript, we noted on this low support values. We re-ran the analysis by reviewing the characters as carefully as possible, and coded the characters that may be affected by ontogenetic changes as “missing (?)” (Lines 347-355). The resulting phylogenetic tree did not differ from the one from previous analysis. While we cannot be conclusive about the phylogenetic position of *Fukuipteryx* considering the low support values, in terms of the significance of the analysis, we have provided a workable hypothesis that there is an unknown clade of basalmost birds (as represented by *Fukuipteryx*) to be tested with additional fossil specimens in the future. Min Wang has performed studies involving osteohistological analyses on Mesozoic birds in the past. We would appreciate it if our paper is reviewed by a reviewer with expertise in osteohistological analyses if our editor feels it is appropriate.

I feel that everything I've written above makes for a straightforward, if not especially high-profile, study. Unfortunately, the manuscript does suffer from editorial problems

throughout—there are numerous spelling errors, and several sentences don't make sense. I do not want to hold the quality of the language in the paper against the authors, but it is clear that the paper will need to be heavily revised in order for it to be clearly understandable by most readers. I have noted a small number of such issues below, as well as other minor comments.

The revised manuscript was reviewed by a native English speaker and grammatical errors are corrected. It is acknowledged in the acknowledgement section.

Line 50: Ornithuromorpha is misspelled

Corrected.

Line 50: The statement about enantiornithines going extinct at the end of the Cretaceous should be supported by a reference

We included a reference.

Lines 53-55: This sentence doesn't make sense as written--I think 'suppressed' is not the right word here?

Corrected.

Description of holotype: What is the difference between 'incomplete' and 'partial'?

To avoid confusion, “incomplete” is used throughout the current manuscript.

Description: The authors frequently use the term 'skeleton' instead of 'bone' or 'element'

Corrected.

Reviewer #2 (Remarks to the Author):

Review of “A new and bizarre basal bird (Theropoda, Avialae) from the Early Cretaceous of Japan”, by Imai and co-authors.

I am happy to know about the discovery of this new and remarkable Cretaceous avian from Japan. Congratulations to the authors for such an interesting discovery, constituting one of the very few preserved three-dimensionally. The preservation is very good and affords valuable anatomical information not only allowing support the validity of the new taxon, but also to discuss evolutionary aspects of early birds as a whole. The high quality images they offer are quite useful to know the osteology of these basal birds. No doubt, the manuscript deserves publication in *Communication Biology*.

Although I have enjoyed by reading the provocative conclusions expressed in the ms, I am not entirely convinced of them.

General considerations about the ms are as follows:

1. The anatomical descriptions are perfect, but I recommend expanding a little bit the universe of comparisons, including other basal birds (particularly pygostilians). Thus, the paper requires more explanations to justify why *Fukuipteryx* falls outside Ornithothoraces, instead of being a basal member of Pygostilia.

We made comparisons of the specimen against Ornithothoraces to contrast the lack of the ornithothoracine characters in *Fukuipteryx* (Lines 303-321).

2. The ms begins with a conclusion: convergence was a common topic among basal birds, and *Fukuipteryx* represents one of these cases. I am not saying that convergence was not present in this part of the tree of life, but I believe the allocation of *Fukuipteryx* outside Pygostilia seems

counterintuitive. It is hard to believe that such a “modern” kind of pygostyle present in *Fukuapteryx* results case of convergence with that of derived pygostilians (e.g., enantiornithes).

Considering that the pygostyle independently evolved within Theropoda (in oviraptorosaurs, therizinosaur, and birds), there is a possibility that the pygostyle evolved independently in an avialan clade basal to *Jeholornis*. Upon receiving the reviewer’s comment, we re-ran the phylogenetic analyses with as much conservative approach as possible in coding character states (Lines 347-355), and the position of *Fukuapteryx* in the most parsimonious tree remained the same. We do not consider that the “modern kind of pygostyle” as noted in the above comment is not present in *Fukuapteryx*. “Modern kind of pygostyle” is compact and ploughshare-shaped seen in ornithuromorphs (including modern birds), while the pygostyle in *Fukuapteryx* is different in being large and rod-shaped. Additionally, individual caudal vertebrae to form the pygostyle are completely fused beyond distinction of individual elements in the derived-avialan ornithuromorphs and enantiornithes, while the pygostyle of *Fukuapteryx* bears incipient neural spines and transverse processes from which individual caudal vertebrae can be discerned, features shared with basal avialan confuciusornithids and *Sapeornis*. These descriptions are given in lines 171-181.

3. Although I guess that *Fukuapteryx* is a pygostilian, I agree with the authors that it does not associate immediately with any of the currently recognized basal pygostilian clades (i.e., Confuciusornithidae, Sapeornithidae, Enantiornithes). Thus, this is the reason I encourage the authors to review their original interpretations, but also considering the possibility that *Fukuapteryx* represents a pygostilian not particularly related with any of the currently recognized clades.

We agree with the reviewer that *Fukuapteryx* constitutes its own clade that was unknown previously. Our interpretation has to remain similar to the previous one, because the result of our revised phylogenetic analysis did not change from the previous one. We realize the incompleteness of the specimen may contribute to the enigmatic position of *Fukuapteryx* within the Avialae, and it is noted in the manuscript. We would like to emphasize that, while we do not intend to persist our phylogenetic interpretation of the specimen, it is currently the most possible hypothesis with the data we are given by the specimen and previous studies. We hope that additional data on these very basal clades of avialans be provided in the future and enigmatic position of *Fukuapteryx* outside of the Pygostilia be clarified.

Particular observations on the ms are as follows:

ABSTRACT

I recommend modifying the text, by almost repeating what it is said in the main text about that early Cretaceous birds "are currently known from the two-dimensionally-preserved specimens from north-eastern China, Las Hoyas (Spain), and Araripe (NE Brazil)". Current knowledge on basal birds does not restrict to China alone.

Due to the restriction in number of words in the abstract, we were unable to expand the sentence as suggested. However, replaced the word “exclusively from” to “largely from” to avoid excluding other localities.

DIAGNOSIS

With the aim to quickly realize the size of the specimen, I suggest comparison with a living bird. For example: "A non-ornithothoracine bird the size of a sparrow/chicken/turkey..." (choose the correct one).

Based on our skeletal reconstruction, the specimen is similar to a pigeon in size, and we included it in the revised manuscript (Line 111).

Lines 122-123 – Authors cite *Anchiornis* as a “non-avian theropod”. However, different recent papers (Agnolín and Novas, 2013; Agnolín et al., 2019; Pei et al. 2018) demonstrated the avian affiliation of *Anchiornis*, as originally proposed by Xu (2011).

We realize that different authors place *Anchiornis* in different positions of the maniraptoran phylogeny. Even in the most recent phylogenetic analysis by Xu et al. (2019), *Anchiornis* falls outside of “Mesozoic avians” and its position is currently under debate. To be fair, we replaced the term “non-avian theropod” with a more inclusive cladistic name “paravian” when referring to *Anchiornis*. All previous studies agree on the paravian affinity of *Anchiornis*.

Lines 137-138 – Authors say that dorsal centrum of *Fukuipteryx* bears large and deep, suboval lateral excavations, as in *Confuciusornis*, but not in more stemward avialans such as *Jeholornis* and *Archaeopteryx*. Important is to also include that enantiornithes also bear such large excavation on dorsal centra.

Zhou and Zhang (2003), ref. 30, makes a statement that all enantiornithes bear this feature, and it is therefore cited in the manuscript (Line 139).

Lines 152-155 - Please, insert information about the number of free caudals in enantiornithes and basal ornithurines.

The above information is included in the manuscript with references (Lines 158-160).

Lines 152-155 – Why do you assume *Fukuipteryx* had more than five free caudals?

We do not assume the animal has more than five, but we suggest it is possible there were additional free caudals when it was alive, as most basal pygostylians have more than five of them. This was made clear in the revised manuscript (Lines 158-160).

Lines 176 – 178 - Similarities that *Fukuipteryx* shares with some enantiornithes in the particular shape of pygostyle, can be used in support that *Fukuipteryx* is a bird related with enantiornithes, rather than consider these features as convergent acquisitions.

As discussed in the manuscript (Lines 303-321), the furcula, coracoid, and metacarpals do not exhibit enantiornithine features. *Confuciusornithids* also share a relatively-large, rod-like pygostyle with *Fukuipteryx*. These lines of evidence support that *Fukuipteryx* is closer to non-ornithothoracine birds than to any enantiornithines.

Lines 183-184 – Authors say “Thus, such feature may have been already present in basalmost birds with pygostyle as suggested in that of *F. prima*”. I suggest to change this phrase for "Thus, such feature may have been already present in basalmost pygostilians as suggested in that of *F.prima*".

The sentence was rephrased as suggested (Lines 188-189).

DISCUSSION

Ideas developed in present "Discussion" are based on the assumption that *Fukuapteryx* is an avialan less derived than *Jeholornis*. However, *Fukuapteryx* exhibits several enantiornithine/ornithothoracine features suggesting a higher position among birds.

Line 292 – Authors say “The following morphological features indicate *Fukuapteryx prima* is an avialan”. However, *Fukuapteryx* exhibits features that are clearly MORE DERIVED than basal avialans (e.g., *Archaeopteryx*, *Jeholornis*, *Rahonavis*).

With the above sentence, we simply argued that *F. prima* is an avialan but not basal paravians (i.e., dromaeosaurids, troodontids and other taxa that belong to Paraves but not to Avialae). We do not argue here whether *Fukuapteryx* is more derived than *Archaeopteryx*, *Jeholornis*, nor *Rahonavis*. Such argument is made later in the discussion.

Lines 296-297 – Authors say “Furthermore, *F. prima* exhibits numerous primitive features comparable to non-ornithothoracine birds”. Please, list these features.

These features are listed following the above sentence (Lines 305-308).

Lines 308-309 – Authors say that *Fukuapteryx* “exhibits several primitive features that cannot be explained solely by ontogeny. For example, the pygostyle is large, robust and well-fused”. Nevertheless, this kind of pygostyle gathers a set of clearly derived features, being morphologically far from a primitive condition, as represented by Sapeornithidae and Confuciusornithidae.

In the paragraph, we list the primitive skeletal characters of *Fukuapteryx* that cannot be exclusively ascribed to ontogenetic variation (those characters that can be safely used in the phylogenetic analysis without confusing developmental and evolutionary features). The sentence was rephrased in the revised manuscript. As suggested by the reviewer the pygostyle is comparable to those in Sapeornithidae and Confuciusornithidae. However, it is primitive in comparison to the pygostyle of Ornithothoraces.

Line 314 – Another feature considered by authors as “primitive” for *Fukuapteryx* is pelvic elements unfused each other. But this is an ontogeny-dependent feature, and cannot be taken as indicative for a basal position of *Fukuapteryx* among Avialae.

As discussed in the manuscript, in most Early Cretaceous birds (especially non-ornithothoracines) the pelvic elements do not fuse throughout ontogeny (Lines 340-344). This feature alone does not indicate the basal position of *Fukuapteryx*, but is one of the multiple lines of evidence that the animal is a basal avialan. To be conservative, though, we coded the character states related to pelvic fusions missing (0→?) in the revised phylogenetic analysis.

Lines 325-327 - The strict consensus tree supports the basal position of *Fukuapteryx prima* within Avialae and outside of Ornithothoraces. I highly respect authors for cladistics analysis they did, but the placement of *Fukuapteryx* as a bird less-derived than *Jeholornis* seems counterintuitive. General aspect of each of the available bones of *Fukuapteryx* (particularly the complex shape of the pygostyle) suggests it as a pygostilian bird. I recommend to discuss this point, as well as to

make some comments if it is closer to Enantiornithes or Jinguofortisidae.

The enigmatic position of *Fukuipteryx* within Avialae and its implications are discussed in Lines 362-378. In Lines 310-326, we discuss that *Fukuipteryx* lacks enantiornithine features and is clearly not close to enantiornithes. *Fukuipteryx* shares some features with Jinguofortisidae as already noted in Lines 207-210, 271-273, and 305-306.

Lines 341-343 – Authors state that “skeletal features of *F. prima* demonstrate that the shortening and fusion of caudal vertebrae are not necessarily associated with the flight adaptations”. Authors are right about that the origin of a pygostyle does not necessarily had an univocal relation with flying capabilities. However, and regarding *Fukuipteryx*, I am unable to recognize which are those "skeletal features" which are not related with flight. On the contrary, all the wing bones of *Fukuipteryx* closely resemble those of a flying bird.

As the reviewer says, *Fukuipteryx* possesses limb skeletal features that are comparable to other flight-capable avialans (including *Archaeopteryx*), but not to the extent of enantiornithines or ornithuromorphs. We feel it was a little too much over-interpretation and decided to omit the part for more conservative approach.

Line 348 – Authors defend that the pygostyle evolved independently from flight capabilities. However, the kind of pygostyle present in *Fukuipteryx* seems more derived than that of *Confuciusornis*, resembling that of enantiornithes. A "pygostyle" (that is: fusion of distal caudals) evolved in oviraptorosaurs and pygostilian birds, and may have evolved many times among basal birds. However, a pygostyle with the system of crests and grooves present in *Fukuipteryx* gathers much more characteristics than "fused vertebrae" alone. I am unable to say that such a complex pygostyle IS NOT related with flight control.

We agree with the reviewer that the pygostyle in *Fukuipteryx* may be flight-related. However, we disagree with the reviewer in suggesting the pygostyle of *Fukuipteryx* is “more derived than that of *Confuciusornis*, resembling that of enantiornithes”. According to Wang and O’Connor (2017) in reviewing the pygostyles of Early Cretaceous birds, it is stated that typical enantiornithine pygostyles are well-fused and individual processes cannot be recognized in adults. They also exhibit “paired dorsolateral ridges demarcating a deeply incised dorsal surface.” “Ventrally, the body is very narrow, with a pair of ventrolateral processes”. These features result in the appearance of “X-shaped in proximal view”. In the pygostyle of *Fukuipteryx*, the individual transvers processes and neural spines are visible, no dorsolateral or ventrolateral processes are present, and it is triangle-shaped in proximal view.

Lines 352-353 – Authors refer about that “The discovery of *Fukuipteryx prima* further increases the geological distribution of non-ornithothoracine birds.” But if *Fukuipteryx*, instead, is a basal pygostilian, then this paleobiogeographic conclusion could be incorrect. Accepting *Fukuipteryx* as a member either of Enantiornithes or Ornithothoraces, then the paleobiogeographic conclusion results more congruent with already known pygostilian/ornithothoracine temporal and paleobiogeographic distributions.

As already discussed, *Fukuipteryx* lacks ornithothoracine features. It still extends the geographical distribution of the non-ornithothoracine pygostilians to further East of Asia.

ILLUSTRATIONS

The illustrations of the present ms are excellent and I hope they become fully published. I have inserted some observations close to some of the illustrated bones. By observing them, it seems clear that *Fukuipteryx* (even accepting it is a pygostilian), it does not comfortably fit among enantiornithes, for example. Thus, although I do not share author's results depicting *Fukuipteryx* as a non-ornithothoracine bird, I agree with them in viewing *Fukuipteryx* as a bizarre pygostilian!

Notes on the figures are as follows:

Figure 3b. The well-developed latero-dorsal crests conform a derived character, absent in basal pygostilians, but present in enantiornithines. This view is almost identical to *Mirarce* (Atterholt et al., 2018).

Latero-dorsal crests in the pygostyle of enantiornithines including *Mirarce* is absent in *Fukuipteryx*. We guess what the reviewer mentions here is the last free caudal vertebra of *Fukuipteryx* that appears X-shaped in the cranial view. However, these caudal vertebrae are not fused with the pygostyle, resulting in the triangular-shape of the element.

Figure 4b. The furcula is robust, but different from enantiornithes in lacking a hypocleidium.

We agree with the reviewer, and this supports our interpretation that *Fukuipteryx* is not enantiornithine.

Figure 4c. The acromial process of coracoid does not surpass the cranial level of coracoid, similar to basal birds and different from derived ornithothoracines.

We agree with the reviewer.

Figure 4c. The coracoid exhibits a modern aspect in being strut-like, different from most basal birds. Thus, it fits well among derived pygostilians. However, it differs from enantiornithines in the following features: 1) proximal end is robust, apparently different from more slender one of enantiornithes; 2) the glenoid is CONCAVE, different from the flat or convex one of enantiornithes; 3) the dorsal surface of coracoid seems devoid of the typical excavation of enantiornithes.

Strut-like coracoid is also found in *Jeholornis*, a basalmost Early Cretaceous avialan. Thus, the strut-like coracoid, while widespread among modern birds, can be present in much more basal birds. We agree with the reviewer that the coracoid lacks typical enantiornithine features, especially the convex glenoid, which is a diagnostic character of the group.

Figure 5a. The hole present on the proximo-medial corner of humerus, is it an artifact of preservation? How it relates with the broken proximal half of deltopectoral crest?

It does not appear to be an artifact of preservation. The inner surface of the "hole" is smooth and does not exhibit any inner-bone texture that would appear if the bone is "excavated" during the preservation. The deltopectoral crest presumably terminates at the "hole" because the proximal corner of the "hole" exhibits smooth surface indicating the crest did not extend beyond it.

Figure 5b. This extensive and rectangular deltopectoral crest is also present among enantiornithes.

It seems that the rectangular deltopectoral crest is common among non-ornithothoracines (Wang et al. 2016) and enantiornithes as suggested by the reviewer.

Figure 5c. Could you offer information if an interosseous groove – a characteristic of many enantiornithes- is present or not on the radius?

No, we did not observe interosseous groove on the radius.

Figure 5d. The abbreviation “imt” is not explained in figure text.

Corrected.

Figure 5. A reconstruction of the entire manus will be welcome!

We do not have a complete series of manual phalanges, even if we combine left and right elements. Thus, we hesitate to create the reconstruction and put it on the manuscript.

Figure 6a. The proximal end of femur reminds me that of El Brete (Argentina) enantiornithes.

The proximal end of the femur also resembles that of Confuciusornithids and Chongmingia. With the lack of the apparent diagnostic features for Enantiornithes (such as convex glenoid of the coracoid) in *Fukuipteryx*, we can only say that enantiornithine-like features the reviewer raises are not indicative of *Fukuipteryx* being an enantiornithine.

Figure 6b. The apparent lack of cnemial crest in proximal tibia resembles enantiornithes.

While not as conspicuous as it is in enantiornithes, cnemial crest is figured in Fig. 6b.

Figure 8. Take note of the spelling of “Ornithuromorpha”

Corrected.

Figure 9b. I suggest removing the life restoration of *Fukuipteryx*, thus gaining space for an informative anatomical illustration depicting, for example, bony proportions in both side and dorsal views.

We replaced the figure with left-lateral, right-lateral, and dorsal views of the skeleton. We feel it is worthwhile to include the life restoration of the animal to give readers (especially those that are not particularly specialized with the study of fossil birds) general idea what it looked like and in what ways it was different from other birds (modern and fossil). Therefore, we keep the restoration in Fig. 9 in the revised manuscript. We are open to removing it if our editor considers the additional use of the space in the manuscript in such purpose is unfavorable.

Reviewers' comments:

Reviewer #1 (Remarks to the Author):

I was impressed with the authors' efforts to accommodate the first round recommendations of both reviewers. I think the new version of the paper is much improved in its clarity. I also think the descriptions are more detailed, and identification of certain problematic elements better justified.

I do still have one question about the surangular. Because this is such a tricky element to identify I think some additional comparisons would be useful. I know the surangular has been figured in some crownward avialans like *Ichthyornis* and it would be useful to know how it compares with that of *Fukuipteryx*.

I will be interested to know what the other reviewer thinks of the revised phylogenetic interpretation as I believe the reviewer raised some important points in their review, but overall I think the authors have effectively responded to my suggestions. I do believe that the paper will need some additional editorial work for optimal readability, but hopefully that is something the journal can assist with.

A few minor corrections (not an exhaustive list):

Line 137: Should be 'heterocoelous'

Line 127: Change to 'are reminiscent of'

Line 121: Change to 'vertebrae'

Line 85: Delete 'of'

Line 60: Check spelling of the family name and reconcile throughout manuscript

Line 44-45: I find the statement about the phylogenetic position of *Jeholornithiformes* to be phrased in a confusing way, because as written it seems to conflict with the definition of *Avialae* provided in the preceding sentence.

Line 41: *Solnhofen* is misspelled.

Reviewer #4 (Remarks to the Author):

Authors have responded each of the observations, comments and suggestions I made on their original ms. They defended their arguments reasonably, and the results they have got after review seem, now, better supported. However, I believe that alternative phylogenetic interpretations will be presented after publication of present ms.

I will be happy to see this contribution published in *Communications Biology* in the near future.

Fernando Novas

Reviewers' comments:

Reviewer #3 (Remarks to the Author):

I was impressed with the authors' efforts to accommodate the first round recommendations of both reviewers. I think the new version of the paper is much improved in its clarity. I also think the descriptions are more detailed, and identification of certain problematic elements better justified.

Reply: We appreciate Reviewer #3's comments upon our initial submission, which improved our manuscript greatly to the current version.

I do still have one question about the surangular. Because this is such a tricky element to identify I think some additional comparisons would be useful. I know the surangular has been figured in some crownward avialans like *Ichthyornis* and it would be useful to know how it compares with that of *Fukuipteryx*.

Reply: We agree that the surangular is difficult to identify as it is not often preserved well in fossil birds, and providing additional comparisons is meaningful. We are hesitant to include overwhelming amount of the detailed anatomy and comparisons on the element in the main text and, instead, these are included in the supplementary file as Supplementary Note 2.

I will be interested to know what the other reviewer thinks of the revised phylogenetic interpretation as I believe the reviewer raised some important points in their review, but overall I think the authors have effectively responded to my suggestions. I do believe that the paper will need some additional editorial work for optimal readability, but hopefully that is something the journal can assist with.

A few minor corrections (not an exhaustive list):

Line 137: Should be 'heterocoelous'

Line 127: Change to 'are reminiscent of'

Line 121: Change to 'vertebrae'

Line 85: Delete 'of'

Line 60: Check spelling of the family name and reconcile throughout manuscript

Line 44-45: I find the statement about the phylogenetic position of Jeholornithiformes to be phrased in a confusing way, because as written it seems to conflict with the definition of Avialae provided in the preceding sentence.

Line 41: Solnhofen is misspelled.

Reply: These were corrected and highlighted in the revised manuscript. We also checked and reconciled the family names throughout the manuscript.

Reviewer #4 (Remarks to the Author):

Authors have responded each of the observations, comments and suggestions I made on their original ms. They defended their arguments reasonably, and the results they have got after review seem, now, better supported. However, I believe that alternative phylogenetic interpretations will be presented after publication of present ms.

I will be happy to see this contribution published in *Communications Biology* in the near future.

Fernando Novas

Reply: We deeply appreciate positive comments and suggestions provided by Reviewer #4, which greatly improved our manuscript. We agree that alternative phylogenetic interpretations will be most likely presented, and believe *Fukuipteryx prima* is in fact important to facilitate such vigorous tests for the phylogenetic relationship of basalmost birds in the future.